# Koshu GRoup Activity, Active Play and Exercise (GRAPE) Study: A Cluster Randomised Controlled Trial Protocol of a School-Based Intervention among Japanese Children

**DOI:** 10.3390/ijerph18073351

**Published:** 2021-03-24

**Authors:** Mitsuya Yamakita, Daisuke Ando, Hayato Sugita, Yuka Akiyama, Miri Sato, Hiroshi Yokomichi, Kaori Yamaguchi, Zentaro Yamagata

**Affiliations:** 1Faculty of Nursing, Yamanashi Prefectural University, Kofu, Yamanashi 400-0062, Japan; 2Department of Health Sciences, Basic Science for Clinical Medicine, Division of Medicine, Graduate School Department of Interdisciplinary Research, University of Yamanashi, Chuo, Yamanashi 409-3898, Japan; yukaa@yamanashi.ac.jp (Y.A.); miris@yamanashi.ac.jp (M.S.); hyokomichi@yamanashi.ac.jp (H.Y.); zenymgt@yamanashi.ac.jp (Z.Y.); 3Division of Human Sciences, Faculty of Education, Graduate School Department of Interdisciplinary Research, University of Yamanashi, Kofu, Yamanashi 400-8510, Japan; dandoh@yamanashi.ac.jp (D.A.); hayato0925@outlook.jp (H.S.); 4Faculty of Health and Sport Sciences, University of Tsukuba, Tsukuba, Ibaraki 305-8577, Japan; yamaguchi.kaori.gf@u.tsukuba.ac.jp; 5Center for Birth Cohort Studies, Graduate School Department of Interdisciplinary Research, University of Yamanashi, Chuo, Yamanashi 409-3898, Japan

**Keywords:** physical activity, sedentary behaviour, wrist-worn activity trackers, bone mass, cluster randomised controlled trial, school-aged children

## Abstract

School-based programmes need to be effective, easy for all, easy to perform within a short duration, and inexpensive. However, no studies have reported whether voluntarily and very short-time active play programmes contribute to improved health outcomes. This study aims to describe the GRoup activity, Active Play and Exercise (GRAPE) cluster randomised controlled trial that examined whether active play interventions of very short durations contribute to increasing physical activity (PA) and bone mass among school-aged children. The trial was conducted in 2018 from January to June, and the activity comprised ≥2 children jumping together for approximately 10 s per session, at least five times a day (approximately 1 min/day). School clusters, pair-matched as per school size (total number of children) and region, were randomly allocated to either intervention or wait-list control groups. The primary outcomes comprised objectively measured changes in PA levels (moderate-to-vigorous PA) evaluated using wrist-worn activity trackers from baseline to the one-year follow-up (six-month post-intervention follow-up) and changes in bone mass evaluated using calcaneus quantitative ultrasound parameters. This study could describe the problems and challenges in school-based PA intervention studies and present findings that could make a potentially important contribution to health education and PA promotion.

## 1. Introduction

Physical activity (PA) is associated with numerous health benefits across an individual’s lifespan [1,2]. Particularly, PA in childhood is one of the most effective preventive strategies to fight osteoporosis later in life [3]. Evidence also indicates that PA is associated with better mental health among children and adolescents [4]. In addition to improving physical and mental health, PA in childhood also promotes certain aspects of social health, such as non-cognitive skills [5] and labour market outcomes in later life [6,7]. Despite these benefits, globally, 81% of children and adolescents aged 11–17 years do not meet the recommended guideline of at least 60 min of moderate-to-vigorous PA (MVPA) daily [8,9].

Among Japanese children, 51.4% of the boys and 70.0% of the girls in fifth grade (aged 10–11 years) engage in exercise for less than 420 min/week, excluding physical education classes [10]. Additionally, owing to the prominent segregation of the time spent in PA or sports participation [11], there is a concern that children who are not good at and/or dislike exercise may have fewer opportunities to participate in exercise or sports. Therefore, the Japan Sports Agency (JSA) and the Ministry of Education, Culture, Sport, Science and Technology-Japan (MEXT) conduct yearly national surveys of physical fitness and lifestyle habits, including PA, and support initiatives derived from the detailed analyses of these surveys to improve physical fitness in Japan. Moreover, the JSA and MEXT spread awareness of the importance of physical fitness among children, providing them with opportunities to familiarise with exercise and sports, and nurture qualities and abilities needed to enjoy sports throughout their lives [12,13].

Present evidence suggests that school-based exercise programmes improve various health indicators among children [14,15,16,17,18]. However, implementing additional programmes may not be feasible in Japanese schools since the regular curriculum is on a tight schedule, requiring several additional tasks to be performed [19]. Furthermore, most evaluations of any feasible, implemented, school-based PA promotion programmes comprised one-arm comparisons (before and after) wherein the effects relative to the control group were insufficiently explored [10]. Therefore, school-based programmes need to be effective, easy for all, easy to perform within a short duration, and inexpensive. The “Bounce at the Bell” programme can be easily implemented by elementary school teachers within a few minutes (~3 min) each day, enhancing bone mass in children [20]. However, there is no evidence of the effect of exercising for less than 3 min a day on a child’s bone mass and other health factors. Moreover, there is strong evidence that supervised exercise, such as physical education, is one of the effective ways to increase PA [21]. However, it is difficult for participants to develop an exercise habit and keep it up when the intervention stops; therefore, long-term effects of interventions are questionable [18]. Health education at school, therefore, requires designing of educational methods and strategies (intervention) so that children can voluntarily (or self-determined) rather than compulsorily enjoy physical activities, and this will successfully result in a physically active lifestyle [22].

The Koshu Group Activity, Active Play and Exercise (GRAPE) study is a two-arm cluster randomised controlled trial (RCT) conducted in elementary schools and with wait-list control groups to examine the effects of active play programmes designed to address above-stated problems. The purpose of the Koshu GRAPE study is to clarify the effect of an active play programme that any child can voluntarily perform quickly and easily, without supervision, and to test the hypothesis that such programmes improve physical (PA and bone mass), mental (depressive symptoms), and social (non-cognitive skills, physical competence and school connectedness etc.) factors. This paper reports the rationale and study design of the GRAPE study. To the best of our knowledge, no studies have yet investigated the effect of programmes with very short durations (10 s per session, 1 min per day) of active play in elementary school children.

## 2. Materials and Methods

### 2.1. Study Design and Setting

The GRAPE study is a school-based two-arm (1:1) cluster randomised and wait-list controlled trial that was designed to improve PA and bone mass and reduce sedentary behaviour among children. It was conducted between December 2018 and December 2019 and targeted all 13 municipal public elementary schools in Koshu City, a rural area in the Yamanashi prefecture of Japan. Presently, there is an ongoing community-based prospective birth cohort study, namely Project Koshu, being conducted in the city since 1988. Under Project Koshu, all expectant mothers who responded to a survey during the obligatory visit to the city office for pregnancy registration were recruited, and their children were followed-up until graduation from junior high school [23].

### 2.2. School Recruitment

All elementary schools in Koshu City were invited to participate in the study (Figure 1). Eligible public elementary schools (n = 13) received a brief explanation regarding the GRAPE study during a conference wherein all the principals of Koshu City schools were gathered. At a later date, the principal and school nurse (yogo-kyoyu) at each school received a detailed presentation about the GRAPE intervention, the study’s evaluation methods, and requirements of involvement. Thereafter, the principal was asked to return a signed consent form at a later day if the school was interested in participating in the study.

### 2.3. Participant Recruitment

Following school recruitment, eligible children within each school were invited to participate in the study. Fourth and fifth grade (aged 9–11 years) boys and girls who did not have any medical conditions that prohibited their participation in active play programmes were eligible. This age group was selected because the children therein were capable of responding to questionnaires, could independently handle accelerometers with minimal assistance, and could be followed up for one year while being in elementary school. Potential participants were excluded if they were absent from school for long-term for reasons such as depression.

### 2.4. Interventions

#### 2.4.1. Active Play Programme

An active play programme was conducted in each elementary school for six months (from 15 January 2019 to 28 June 2019). The intervention required the children (≥2) to jump for approximately 10 s per session, at least five times a day (approximately 1 min/day). These short periods of activity were designed to be easy and enjoyable for everyone. Excluding the spring break, the intervention period lasted for 21 weeks. The participants performed three types of activities during the first week, with a new type of activity added each week to prevent the children from getting bored, providing a total of 24 activity types. Additionally, the activities were to involve any of the elements of the four categories of games distinguished by Caillois: Agon (competition), Alea (luck/chance), Mimicry (mimesis), and Ilinx (vertigo) [24]. To explain each activity, an instruction manual sheet (Figure 2), describing the method and illustration, was placed in each classroom. Moreover, a miniature version of the same instruction manual was distributed to each child by the homeroom teacher, and each child understood how to perform the activity themselves. However, since this programme aimed at encouraging children to play spontaneously, the classroom teacher was informed not to enforce an activity if the children had not implemented it. 

#### 2.4.2. Challenge Card

Additionally, a record card was used, which utilised human behavioural characteristics based on behavioural economics to encourage children to enjoy the activities continuously. Specifically, a challenge card that adopted a rank-up/down system that triggered esteem needs and loss aversion (a cognitive bias that refers to the human tendency to prefer avoiding losses to acquire equivalent gains) in addition to a visualisation of results was used (Figure 2). To confirm the rank, children attached their current rank sticker to the challenge card. If children achieved their rank-wise weekly goals, their ranks were increased the next week (rank order: seeds → growth → silver → gold → star (or heart) → diamond), and if they could not achieve their goals, their next week’s ranks were decreased. Children who conducted the activities together were instructed to record each other.

#### 2.4.3. Watching Videos

In order to maintain long-term awareness of active play in the GPAPE programme even after the completion of the intervention, participants of the intervention group watched an approximately 2-min-long movie with the message that physical activities such as standing, walking, playing with friends, etc., can be definite exercise even if it does not involve strenuous activities (https://youtu.be/dcr8aUpd3wk, accessed on 19 March 2021).

#### 2.4.4. Wait-List Control Group

During the intervention period, the participating school children belonging to the wait-list control group continued their routine school life and only underwent evaluation similar to the intervention group. After the final evaluation in December 2019, the wait-list control group received an activity programme similar to the intervention group but more selective and compact (intervention period: three months).

### 2.5. Outcome Measures

#### 2.5.1. Primary Outcome

The primary outcomes of the study were the objectively measured changes in PA level (moderate-to-vigorous PA, MVPA) from baseline to one-year follow-up (six-month post-intervention follow-up) and post-intervention changes in bone mass indicated, measured using quantitative ultrasound parameters (e.g., Stiffness index calculated from speed of sound (SOS) and broadband ultrasound attenuation (BUA).

#### 2.5.2. Secondary Outcomes

The secondary outcomes included objectively measured changes in MVPA, immediately after the intervention; changes in steps, light PA (LPA), sedentary behaviour, and proportion of engagement in the recommended hour of PA (60 min) per day; changes in self-reported MVPA and sedentary behaviour; mental and social health variables immediately and at six months post-intervention; changes in the above variables in subgroups of children disliking exercise or in a being precontemplation stage of the transtheoretical model of behaviour change.

### 2.6. Data Collection

Objectively measured PA and questionnaire data were collected at four time-points: at baseline; a four-week follow-up (mid-intervention); a 6-month follow-up (immediately post-intervention); and a one-year follow-up (six-month post-intervention). Bone mass data were collected at two time-points: during September 2018 as a baseline measure and during September 2019 as a follow-up measure; both collections coincided with the health check-up routinely conducted at the beginning of the second semester (Table 1).

#### 2.6.1. Objectively-Measured Physical Activity

Objectively measured PA was evaluated by Fitbit Ace (Fitbit, Inc., San Francisco, CA, USA), a wrist-worn activity tracker for children aged eight years and older. It features a tri-axial accelerometer and continuously acquires data. It has onboard storage capacity for approximately five days of data without syncing. Participants were asked to wear the device on the non-dominant wrist for 24 h and for 10 days during each of the four time-points, except while bathing, swimming, or playing sports with a high risk of injury. Additionally, participants were asked to charge the Fitbit Ace’s battery during the first period in the classroom every day. Moreover, using Bluetooth technology, the investigators synced the Fitbit’s mobile app data to a Fitbit server two or three times (every third or fourth day) during the 10 days of each period. These collected data from Fitbit Ace devices were downloadable via an application programme interface (API), and the data collected at 1 min intervals provided information on step counts and the time spent in the following activity levels: sleep, sedentary, light, fairly (or moderately) active, and very active/vigorous). Although the Fitbit Ace device has not been established as a valid objective measure of PA yet, similar wrist-worn Fitbit devices (Fitbit Flex2, Fitbit, Inc., San Francisco, CA, USA) have shown excellent correlations between step counts and ActiGraph GT3X+ (ActiGraph, Pensacola, USA), the most valid and widely used tool to measure PA in research [25]. Fitbit Charge HR (Fitbit, Inc., San Francisco, CA, USA) also has adequate sensitivity in classifying moderate and vigorous activity and sleep [26].

#### 2.6.2. Self-Reported Physical Activity and Sedentary Behaviour

Self-reported PA was measured using the following three questions used in the Health Behaviour in School-Aged Children study: (1) Number of days with MVPA ≥60 min (days/week); (2) frequency of VPA (days/week); and (3) duration of VPA (hours/week) [27]. The validity and reliability of these items are acceptable among children [28].

Sedentary behaviour, such as screen time while watching TV or DVDs, video gaming, and using smartphones and PCs, etc., were measured separately on weekdays and holidays. Additionally, data regarding PA-related variables, such as participation in organised sports and active school transportation, were collected.

#### 2.6.3. Bone Mass

A bone mass survey for children in elementary schools from grade 4 to junior high school grade 3 (aged 9–15 years) has been conducted every August and September since 2006 as part of the Project Koshu. Bone stiffness was assessed as an indicator of bone mass using quantitative ultrasound (QUS). QUS measurements were performed with an Achilles A-1000 Express II (GE Healthcare, Milwaukee, WI, USA). This portable device measures bone stiffness using ultrasound waves and presents three parameters: (1) broadband ultrasound attenuation (BUA) which reflects the absorption of sound waves (dB/MHz); (2) SOS which expresses the stiffness of a material using the ratio of the traversed distance to the transit time (m/s); (3) stiffness index (SI) which is an automatically calculated parameter that combines BUA and SOS values [SI = (0.667 × BUA) + (0.278 × SOS) − 417]. The SI of the right foot calcaneus was used as the bone mass parameter in this study. To equalise the measurement conditions, setup was completed 20 min before the start of measurement for children; the room temperature at the time of measurement in each school was controlled at 25–27 °C, calibration was performed, and the stability of the measurements were confirmed.

#### 2.6.4. Bone Mass-Related Measures

Age in months, body composition measurements (body height, weight, and body mass index), calcium intake, puberty status, and socioeconomic status were identified as bone mass-related measures. Detailed evaluation for these variables is previously described [29].

#### 2.6.5. Mental Health

As an index of mental health, the rate of depressive symptoms evaluated by the Japanese-translated version of Birleson Depression Self-Rating Scale for Children (DSRS-C) [30,31], which is a valid and reliable tool. The questionnaire was obtained from another survey conducted every July annually under Project Koshu. DSRS-C consists of 18 questions with responses including ‘most’, ‘sometimes’, and ‘never’ scored as 2, 1, and 0, respectively. The total score was calculated as the DSRS-C score. Based on previous studies, a score of 16 or higher indicated depressive symptoms. The evaluation in July 2018 was used as the baseline data, and the evaluation in July 2019 was used as the post-intervention follow-up result.

#### 2.6.6. Social Health

The GRAPE study questionnaires included the following social health variables: attitude to PA, attitude to physical education (PE), behaviour change stage for PA, physical competence, school connectedness, personality traits, non-cognitive skills (grit, self-esteem, and self-control). 

The children’s attitude to PA and PE was determined using four responses (like, a little like, a little dislike, dislike) when asked whether they liked or disliked PA and PE classes. The behaviour change stage was assessed using the Transtheoretical Model of Change (TTM) proposed by Prochaska and Velicer [32]. Depending on the actual behaviour in the past and present and the readiness for that behaviour, the behaviour was divided into five stages: ‘precontemplation’, ‘contemplation’, ‘preparation’, ‘action’ and ‘maintenance’. Physical competence was evaluated using the Physical Competence Scale developed by Okazawa et al. [33]. This scale used 12 items to evaluate three factors: ‘physical competence’ (confidence to do physical exercise well); ‘feeling of control’ (confidence to do physical exercise well through effort and training), and ‘peer and teacher acceptance’ (recognition accepted by peers and teachers). School connectedness was measured using three items: ‘I like school’, ‘School is a nice place to be’, and ‘Sense of belonging at school’; the first item was answered using a four-point scale (like very much, like, dislike a little, dislike a lot) and the second two items were answered using five options (strongly agree, agree, neither agree or disagree, disagree, strongly disagree) [34]. The items were summed to create a composite score. Personality traits were assessed using a valid and reliable Japanese version of the Ten Item Personality Inventory (TIPI-J) [35]. The TIPI-J is a scale of the Big-Five personality dimensions: Extraversion, Conscientiousness, Agreeableness, Neuroticism, and Openness to experience. Each item was scored on a 1–7-point scale, and the scores were summed for each of the 5 dimensions.

##### Non-Cognitive Skills

As non-cognitive skills, grit, self-esteem and self-control were measured. Grit was assessed using the Japanese-translated version of the eight-item grit scale for children developed by Duckworth et al. [36]; the internal validity of the scale is confirmed by the authors [37]. The average value of the five options for the eight items was determined as a grit score (5: extreme grit, 1: lack of grit). Self-esteem was estimated using the Two-Item Self-Esteem scale (TISE). The TISE consists of two aspects of the self-esteem concept (self-evaluation and self-acceptance) and demonstrated concurrent validity by positive high correlations between the TISE and pre-existing self-esteem scales (i.e., Rosenberg’s self-esteem scale [38,39]). Self-control was evaluated by the Japanese-translated version of the Brief Self-Control Scale (BSCS-J) [40]. BSCS is a valid and reliable scale consisting of 13 items with responses that include five options ranging from ‘very much like me’ to ‘not like me at all’. As a parameter of family socioeconomic status, family affluence was measured using the family affluence scale, which has been validated in the European Health Behaviour in School-aged Children study [41] and is based on four questions: family car ownership; number of family trips per year; having one’s own bedroom; and total number of computers at home.

#### 2.6.7. Others

Other questions included undertaking non-sports lessons (such as cram schools; private tuition, yes/no), time spent on studies in a week (hours/week), knowledge of the effects of PA (five questions about the association between exercise, osteoporosis, cancer, memory, income and sedentary behaviour). 

### 2.7. Sample Size

The sample size was calculated based on the effect sizes (d = 0.50) derived from our previous study [29]; for example, the sample size was determined to detect a mean difference of five SI between the intervention and control groups with a standard deviation (SD) of 10 SI, and a mean difference in MVPA of 10 min/day with a SD of 20 min. A total of 63 children were required per group to detect a change in effect size with the alpha level set at 0.05 and power at 0.80.

To account for clustering, the required sample size of 63 children was adjusted by a design effect of 1 + (m − 1) ρ, where m is the average number of children per cluster and ρ is the intra class coefficient (ICC). We estimated that each school had 20 children and an ICC of 0.05. Using these estimates, the total required sample size was found to be 246 children (63 × (1 + (20 − 1) × 0.05 × 2 groups)). Furthermore, we assumed that 70% of the schools and 90% the children would accept participation, and 10% would drop-out during the study period. Therefore, it was assumed that at least 433 children were needed. Therefore, we recruited children from the 4th and 5th grade of all schools in Koshu City (13 schools, 511 children).

### 2.8. Randomisation

Randomisation was conducted at the school level after baseline data were collected. This study was conducted in nine schools (18 classes, one class in each 4th and 5th grade; Figure 1). These schools were matched by size (total number of children) and region (Enzan or Katsunuma area) into four pairs. These pairs and one school with the smallest number of children in Koshu City were randomly assigned to either the intervention group or the control group using random numbers in Microsoft Excel by a researcher who was not directly involved in the implementation. The schools were each assigned a random number, and the school with odd last digits random number in each pair was assigned to the GRAPE intervention group. If the last digit was the same, GRAPE intervention group was assigned by the last second digits random number.

### 2.9. Blinding

Given the nature of the active play programme, it is impossible for teachers to be blinded to group allocation. However, children were not informed about the group differentiations; they were only informed that they would be performing the activity first (intervention group) or later (wait-list control) and, therefore, were blinded to group allocation. In addition, bone mass was measured by research assistants who were unaware of the allocation, PA was assessed using objectively measuring wearable devices, and other outcome data were obtained through self-reporting; thus, reducing potential outcome assessor bias.

### 2.10. Statistical Analysis

For inter-group comparisons of characteristics at baseline, independent t-tests and chi-square tests will be used for continuous and categorical variables, respectively. Missing values will be complemented using the multiple imputation method, which is integrated using Rubin’s formula [42], by creating 20 datasets under the assumption of the missing measurement mechanism of missing at random (intention-to-treat). To evaluate the effect of the intervention programme, a general linear mixed model will be employed for the analysis of continuous variables. Sensitivity analysis will be conducted for complete case analysis excluding data with missing values. Regarding the analysis of dichotomous secondary outcomes, the intervention effect will be examined by comparing the odds ratio estimated using multi-level (logistic model) regression analysis. Additionally, subgroup analysis will be performed in the population disliking exercise or precontemplation stage of transtheoretical model of behaviour change. All statistical analyses will be performed using Stata SE version 15.1 (StataCorp, Spring, TX, USA).

### 2.11. Dissemination

The results of the study will be published in conferences and peer-reviewed science journals, and booklets or posters summarising the results will be returned to the participating schools. Additionally, the results of this research will be directly returned to the children, guardians, schools and local residents by reporting at the School Health Committee and creating posters or booklets.

## 3. Discussion

This GRAPE study programme is the first study to examine whether voluntary and a very short active play program contributes to increased bone mass and PA in school-aged children. The study is expected to improve the participant’s health indicators, such as increased physical activity and bone mass, mental and social health, and non-cognitive skills, based on the hypothesis that even a small amount of time spent on active play involving jumping will show positive effects on daily repetition. Additionally, the results of this study will help to develop evidence-based effective exercise programmes and health education in schools wherein the effects relative to that in a control group were insufficiently explored. Moreover, by objectively evaluating physical activity using a consumer model wrist-worn accelerometer, it will be possible to validate results, and children will become aware of their own levels of physical activity. However, the interpretation of GRAPE study results will likely be limited owing to the lack of generalizability as the participants were selected from a single rural area. The clusters were small, and there were not enough participants within the clusters, which resulted in a bias between the intervention and control groups. Moreover, as many measures were self-reported by the children, it is possible that measurement errors will occur. Furthermore, because many outcomes will be evaluated, the problem of “multiple comparisons” will need proper attention.

The implementation of this RCT program was possible by building long-term relationships of trust and cooperation between the health promotion section of the city office, the board of education, each school, and researchers through the Koshu project for. Therefore, the findings of this study will be directly returned to the children, guardians, schools and local residents by reporting to the School Health Committee and creating posters or booklets. Thus, this study not only provides novel academic findings but also returns familiar and evidence-based information to the community; therefore, it will be an academically and practically meaningful study.

## 4. Conclusions

Given the present lack of evidence regarding school-based cluster RCTs in Japan, our study protocol could potentially highlight the many considerations and challenges faced in implementing school-based PA intervention studies. Moreover, the findings of this study investigating the effect of programmes with very short durations (10 s per session, 1 min per day) of active play could make important contributions to the promotion of PA and health education for preventing non-communicable diseases in elementary schools.

## Figures and Tables

**Figure 1 ijerph-18-03351-f001:**
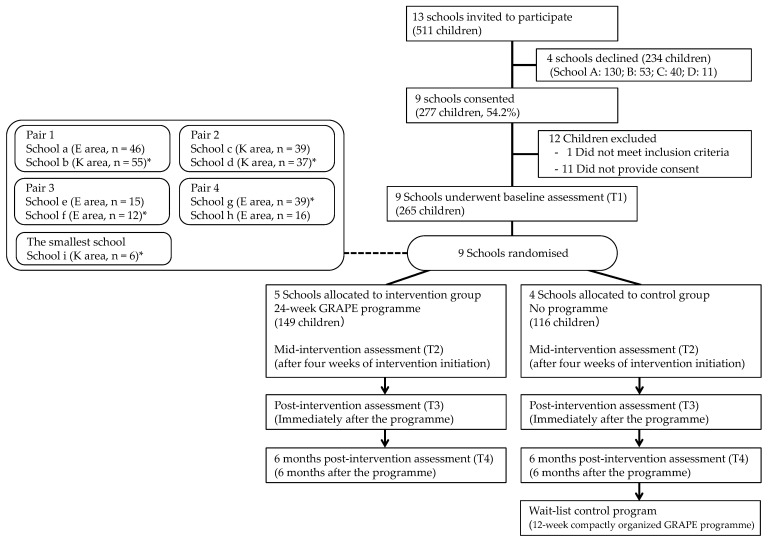
The flow diagram of the study protocol. GRAPE, Group activity, Active Play and Exercise; E area, Enzan area; K area, Katsunuma area; * indicates intervention group.

**Figure 2 ijerph-18-03351-f002:**
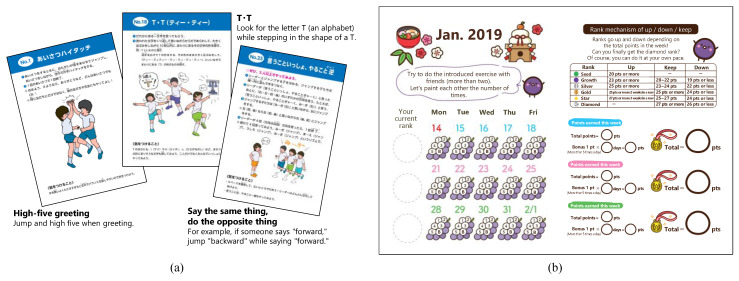
Examples of the instruction manual sheets for activities (**a**) and recording card used in the programme (example month: January) (**b**).

**Table 1 ijerph-18-03351-t001:** Summary of measures among participants in the GRAPE programme.

		T1 December 2018	T2 Febuary 2019	T3 July 2019	T4 December 2019
**PA & Sedentary behaviour**					
	Objectively (Fitbit Ace) mesaured PA and Sedentary behaviour (including sleep)	〇	〇	〇	〇
	Self-reported PA and sedentary behaviour (screen time)	〇	〇	〇	〇
	Active school transportation	〇		〇	〇
**Bone mass**					
	Calcaneus quantitative ultrasound				
**Bone mass-related measures**					
	Anthropometric data (using health check-up data)	〇 ^b^	〇 ^b^	〇 ^b^	〇 ^b^
	Calcium intake, puberty status (only girls)	〇 ^a^		〇 ^a^	
**Mental health**					
	Depression sympton (Birleson Depression Self-Rating Scale for Children)	〇 ^c^		〇 ^c^	
**Social health**					
	Attitude to PA	〇	〇	〇	〇
	Attitude to PE	〇	〇	〇	〇
	Behaviour change stage for PA (Transtheoretical model)	〇	〇	〇	〇
	Physical competence	〇	〇	〇	〇
	School-connectedness	〇	〇	〇	〇
	Personality traits (Ten Item Personality Inventory)	〇	〇	〇	〇
	Non-cognitive skills (Grit, self-esteem, self-control)	〇	〇	〇	〇
	Family socioeconmic status (Family affluence scale)	〇	〇	〇	〇
**Other variables**					
	Non-sports lessons	〇	〇	〇	〇
	Time spent in studying	〇	〇	〇	〇
	Knowledge of the effects of PA	〇	〇	〇	〇

T1 = baseline, T2 = Mid-intervention (after 4 weeks of intervention initiation), T3 = Post-intervention (immediately after intervention initiation), T4 = Follow-up (6 months after intervention initiation); GRAPE; Group Activity, Active Play and Exercise; PA: physical activity; PE: physical education. ^a^ = measured by another survey of Project Koshu conducted in August to September. ^b^ = used from health check-up data conducted at the beginning of the semester (T1 = beginning of first semester (April), T2 = beginning of third semester (January), T3 = beginning of second semester (August to September), and T4 = beginning of third semester (January)). ^c^ = measured by another survey of Project Koshu conducted in every July.

## Data Availability

Not applicable.

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
