# Peer review of "Koshu GRoup Activity, Active Play and Exercise (GRAPE) Study: A Cluster Randomised Controlled Trial Protocol of a School-Based Intervention among Japanese Children"

_ijerph, 2021, doi:10.3390/ijerph18073351_

Round 1
Reviewer 1 Report
The article describes the study based on active play and exercise (GRAPE) that examined whether children from public elementary schools participating in very short-duration active play interventions contribute to increased physical activity (PA) and bone mass, thereby that could have benefits for your health.
Regarding the manuscript, I point out some evaluations:
Abstract and keywords:
The abstract includes the necessary elements: background with purpose (objective) of the study, methods, results, main conclusions without exaggerating them.
- Introduction:
Sufficient ordered references of publications considered as key are indicated, with significant and sufficient evidence. Similarly, reasons that justify the importance in a broad context and the current state of the subject under investigation are highlighted. The study is clearly defined and indicates the intention and meaning of the work.
The hypothesis to be tested in the study is recorded and refers to programs that improve physical health, mental and social factors.
The text is understandable and makes clear the main objective of the work and the main conclusions.
- Materials and Methods:
The GRAPE study is described in detail as a two-arm clinical trial. Also, It is described the Koshu Project that monitors from pregnancies to the completion of children in secondary school.
The information to participate in the GRAPE Study is indicated based on the intervention protocol that is explained to the directors and nurses of the schools regarding the way of evaluating, the methods and requirements for participation. Informed consent is presented by the director of the School upon showing interest in participating in the study. In addition to the methods, the intervention requirements are indicated in sufficient detail through the information meeting held.
Participants
The procedure for the incorporation of the sample participants is indicated, as well as the criteria that must be met: age, medical condition, use of accelerometers and stay in school for one year. Exclusion criteria are indicated if a participant does not have the ability to answer questionnaires, if they did not meet any of the above requirements or due to depression.
Figure 1 satisfactorily clarifies the flow to follow: the invitation to the schools, the knowledge of the elementary schools that resigned, the exclusion of children for not being able to participate due to not meeting the requirements, the consent of participating schools, the method and procedure to continue for the established time and the relationship and process with the experimental group and the control group.
Interventions
The procedure to be followed in the active game program in each elementary school is explained, the time and activities are indicated. In the same way, the instruments that are provided to see in the classroom and use individually. Figure 2 is clarifying.
The “challenge card” instrument motivates participating children to enjoy the activity. An example is indicated how to participate in the activity, analyzing the results that allow you to move up or down the rank. Watching the proposed video helps in the central idea in favor of simple physical activity. All of this manages the flow and interest in participation.
The relationship of the procedures between the experimental group and the control group is conveniently indicated at the different moments of the investigation.
Outcome measures
The way to mediate and analyze the data according to the primary outcomes is indicated from established quantitative parameters of different types, as well as the secondary outcome measures, based on behavioral changes that address different variables.
Data collection
The way of collecting the data in the four moments seems adequate, coinciding with the mid-process medical check-ups. Instruments for collecting data in children, the procedure for storing data that provide information on step count and time spent on sleep activities are noted. Reliability is referred to in the correlation between step count and sensitivity to classify activity and sleep. The validity and reliability of these elements are specified.
The data specification is divided by subtitles, which relate physical activity, sedentary lifestyle, bone mass and the measures in which they are specified, mental health, social health and non-cognitive skills. In each one of them, an adequate and precise description is provided regarding the programs, models, instruments and devices for mediation and their conditions and the parameters for calculating the values with reference to scales, type of items and responses, such as seen in table 1.
Sample size
The sample size with the 511 fourth and fifth grade children from 13 schools in the city of Koshu, seems adequate according to the calculation indicated to detect the changes produced in the application of devices and scales of the program.
- Discussion
It is gratifying to see how the results are going to be disseminated.
It is not so important to emphasize if this is the first study, which may be, but rather that the study provides remarkable academic findings, which are returned to schools and the community, and implies an improvement in physical, social and cognitive health through physical activity with the characteristics indicated.
It would be convenient to specify the verification of the hypothesis raised, in the sense that these applied programs improve physical health (PA and bone mass). In the same way, the strengths and limitations of the study could be highlighted, if any. It is also important to indicate how future research can give continuity to this.
Except for these last observations, I confirm the perceived good level and the correct inclusion of the elements reflected in the manuscript. I congratulate the authors for their contribution to science, for the direct benefit of children.
Author Response
Dear Editor and Reviewers:
We appreciate the reviewers’ polite and valuable comments. Our point-by-point responses to their specific comments are as follows. Additionally, when revising the manuscript, we improved some of the aspects that came to our notice. Specifically, we thoroughly checked and edited the English in the manuscript. The modified parts were highlighted using the "Track Changes" function in Microsoft Word.

Reviewer 2 Report
Real cool study. Good job.
A couple things you should address.
- In the Abstract you write about 'school based programmes' and then later in the abstract you talk about 'active play programs'. This needs to be the same.
- Lines 132 and 150, you use the word 'we'. I think writing in 3rd person would be better. For example, say, 'The researchers used a challenge....
- In you discussion, I think one sentence that generally is positive about the study. For example, 'It was generally found that short bout of jumping were beneficial for students.' Something like that.
Once again, nice study.
Author Response

(The authors gave the same response as above.)

Reviewer 3 Report
The manuscript "Koshu GRoup activity, Active Play, and Exercise (GRAPE) study: A cluster randomised controlled trial protocol of a school-based intervention among Japanese children" is a study protocol for a randomized control trial with two arms looking to evaluate the impact of short term high intensity physical activity in children in Japan.
The manuscript does an excellent work at justifying the need for the study by having a near perfect introduction that justifies the need to evaluate these type of interventions with two arms. At the level of the introduction and abstract, a very minor observation is that authors need to spell out what the RCT acronym stands for in lines 23 and 80.
For the most methods are well written, but a few clarifications are needed. In the flow diagram of Figure 1, the big difference in recruited students (511 to 277) was due to larger schools declining to participate?
The video in line 164 is very interesting, but it will be a more effective resource if subtitled in english for an international audience, especially the part about going at your own pace and enjoying with friends.
In line 318 it is not clear what is the previous study.
It will be nice to see a flow diagram explaining the matching of the schools described in lines 332-342.
It is also very nice the study protocol followed recommendations for RCT from the Equator network.
Author Response

(The authors gave the same response as above.)
